# Evaluating the Diverse Anticancer Effects of Laos *Kaempferia parviflora* (Black Ginger) on Human Melanoma Cell Lines

**DOI:** 10.3390/medicina60081371

**Published:** 2024-08-22

**Authors:** Gyun Seok Park, Juhyun Shin, Seongwoo Hong, Ramesh Kumar Saini, Judy Gopal, Jae-Wook Oh

**Affiliations:** 1Department of Bio-Resources and Food Science, Konkuk University, 120 Neungdong-ro, Gwangjn-gu, Seoul 05029, Republic of Korea; bhs2945@hanmail.net; 2Department of Stem Cell and Regenerative Biotechnology, KIT, Konkuk University, 120 Neungdong-ro, Gwangjn-gu, Seoul 05029, Republic of Korea; junejhs@konkuk.ac.kr (J.S.); hsw2332@naver.com (S.H.); 3School of Health Sciences and Technology, UPES, Dehradun 248007, Uttarakhand, India; saini_1997@yahoo.com; 4Department of Research and Innovation, Saveetha School of Engineering, Saveetha Institute of Medical and Technical Sciences (SIMATS), Chennai 602105, Tamil Nadu, India

**Keywords:** black ginger, anticancer activity, human melanoma, apoptosis

## Abstract

Cancer has become a consistent concern globally and increasingly fatal. Malignant melanoma is a rising concern, with its increased mortality. *Kaempferia parviflora* Wall. ex Baker (*K. parviflora* (KP)), commonly known as black ginger, is well known for its medicinal contributions. For the first time, in the following study we investigated the antimelanoma potential of Laos KP extracts in human cell lines. KP extracts (KPE) in methanol, DCM, and ethyl acetate showed strong cell inhibition in both melanomas, with KPE-DCM being particularly effective in inhibiting melanoma cell migration, invasion, and proliferation by inducing cell cycle arrest and apoptosis, while KPE-Hexane exhibited a low cell inhibition rate and a more limited effect. KPE affected the increased expression of caspase-3, PARP andBax and the decreased expression of the BcL-2, Mu-2-related death-inducing gene (MUDENG, MuD) protein. Furthermore, KPE enhanced apoptotic cells in the absence and presence of the pancaspase inhibitor Z-VAD-FMK. Interestingly, these apoptotic cells were significantly suppressed by the caspase inhibitor. Moreover, elevated mitochondrial membrane potential (MMP) and intracellular reactive oxygen species (ROS) levels, suggestive of KPE’s mitochondrial-mediated apoptosis in melanoma cells, were also confirmed. KPE treatment increased MMP levels, and upregulated the generation of ROS in A375 cells but not in A2058 cells. However, pretreatment with an ROS scavenger (NAC) suppressed KPE-induced cell death and ROS generation. These results clearly pointed out KPE-induced mitochondrial-mediated apoptotic cell death as the mechanism behind the inhibition of the human melanoma cells. Future studies exploring the role of specific ROS sources and their interaction with mitochondrial dynamics could deepen the existing understanding on KPE-induced apoptosis.

## 1. Introduction

*Kaempferia parviflora* Wall. ex Baker (KP), commonly known as Thai black ginger or “Krachai dum” in Thailand [1], is a medicinal plant from the *Zingiberaceae* family. Unlike other ginger varieties, the rhizome of KP is not used in cooking but is instead utilized in traditional oriental medicine for treating allergies, gastrointestinal disorders, fungal infections, and impotence [2]. KP exhibits several pharmacological effects, including anti-allergic [3,4], antimicrobial, anti-inflammatory [5,6,7], and anticancer [1] properties. An extensive review by Chen et al. [8] has highlighted the vascular relaxation and cardioprotective activities of KP along with its benefits in enhancing sexual function, regulating cellular metabolism, providing neuroprotection, and its anti-inflammatory, antioxidant, anticancer, anti-allergic, anti-osteoarthritis, and transdermal permeable properties.

Previous studies using KP crude extracts or purified compounds have been proven for their toxicity against cancer models. In addition, the antimelanogenic activity of 7,4′-dimethylapigenin and 3,5,7-trimethoxyflavone, purified from KPE, were confirmed against mouse-derived melanoma B16F10 cells [9], KP inhibited cervical cancer HeLa [10], ovarian cancer cell SKOV [1], and breast cancer MCF-7 [11]. KP triggers apoptosis through the intrinsic pathway by altering the ratio of Bcl-2 and Bax, along with the activation of Caspases 3 and 9 in gastric cancer SNU-16 cell lines [12]. These existing studies confirm the anticancer activity of KPE. Generally, the spread of cancer cells is facilitated by the production of matrix metalloproteinases 2 and 9, which break down matrix proteins when cancer cells are preparing to metastasize [13].

Malignant melanoma is an aggressive and highly lethal form of skin cancer, accounting for the majority of skin cancer-related deaths despite constituting only 5% of all skin cancer cases [14,15]. Currently, treatments for melanoma include surgical removal, chemotherapy, radiotherapy, and molecular-targeted therapies. However, many patients develop primary or acquired resistance to chemotherapy. Therefore, there is an urgent need to explore new natural therapeutic components to combat melanoma progression and improve patient outcomes [16].

Apoptosis is the most well-understood form of programmed cell death, characterized by specific morphological features such as cell shrinkage, nuclear chromatin condensation, and nuclear fragmentation [9]. The Mu-2-related, death-inducing gene (MUDENG, MuD) protein, which contains a Mu homology domain derived from adaptor proteins, is crucial in intracellular trafficking pathways. Previous research has demonstrated that MuD has anti-apoptotic properties [17].

In this study, we investigated the effects of black ginger extracts on human melanoma cell lines A375 and A2058. While the impact of KP on various cancer cell lines has been documented, its effect on melanoma cell lines has not been previously explored. This study is the first to report that KP extracts induce apoptosis in melanoma cells via mitochondrial damage. The MUDENG perspective behind the KPE anticancer activity has also been highlighted for the first time. Various aspects of the interaction between KP extracts and melanoma cell lines were analyzed and reported. The findings provide compelling evidence that black ginger extracts hold potential as a novel antimelanoma therapy, triggering positive feedback for their therapeutic use.

## 2. Materials and Methods

### 2.1. Extraction and Isolation of KP Extract from the Rhizomes of Black Ginger

Figure 1 shows the overall workflow undertaken in this study. Fresh ginger rhizomes (*Zingiber officinale*) were procured from the local vegetable marketplace of Lak Sao, Khamkeut District, Bolikhamsai Province, Laos. Lyophilized, powdered rhizomes (250 g) of *K. parviflora* were extracted using methanol in a Soxhlet apparatus. The methanol was evaporated under vacuum, and the residue was suspended in water and successively extracted with hexane, dichloromethane, and ethyl acetate. These extracts will be identified as KP extracts (KPE), represented alongside the solvent extract as KPE-DMSO, KPE-MeOH (KPE in methanol), KPE-hexane (KPE in hexane), KPE-DCM (KPE in dichloromethane), and KPE-EtOAc (KPE in ethyl acetate).

### 2.2. Characterization of Methoxyflavones from KPE Using High-Performance Liquid Chromatography (HPLC)

Chromatographic separation was conducted using an Agilent Model 1100 HPLC system (Agilent Technologies Canada Inc., Mississauga, ON, Canada). This system featured an autosampler, dual pump, degasser, diode array detector (DAD), and a Sunfire C18 column (250 × 4.6 mm, 5 μm). The column temperature was set to 20 °C. The mobile phases were 0.1% formic acid in water (phase A) and 0.1% formic acid in acetonitrile (phase B). The gradient elution profile started with 10% B, increasing to 50% B over 30 min, then to 100% B over the next 45 min, followed by a 10-min hold at 100% B and a 5-min post-run, all at a flow rate of 1 mL/min. The samples were scanned across a range of 200 to 800 nm with a response time of 0.05 min (1 s) and a detection wavelength set at 280 nm. We compared the chromatographic peaks to those of standards (5,7-dimethoxyflavone; DMF, 4′,5,7-trimethoxyflavone; TMF, 3,5,7,3,4-pentamethoxyflavone; PMF purchased from iterPharm, Koyang-si, Republic of Korea) and identical retention times were observed for DMF, TMF, and PMF in KP extracts.

### 2.3. Cell Culture and Sample Preparation

Human melanoma cell lines A375 and A2058 were acquired from ATCC (Manassas, VA, USA). Both cell lines were cultured in Dulbecco’s Modified Eagle Medium (DMEM) supplemented with 10% (*v*/*v*) fetal bovine serum and 1× antibiotic (Welgene, Gyeongsan, Republic of Korea). These cell lines served as the primary models for the anticancer assays conducted in this study. KPE extracts (methanol, hexane, dichloromethane, EtoAc) were stored in a deep freezer until use. KPEs dissolved in dimethyl sulfoxide (DMSO) at a concentration of 100 mg/mL were used for in vitro biological tests.

### 2.4. Cell Viability Assay

Melanoma cells were plated at a density of 1 × 10^4^ cells per well in a 96-well microplate and incubated overnight at 37 °C with 5% CO_2_. The following day, cells were exposed to varying concentrations of solvent extracts (3.12, 6.26, 12.5, 25, and 50 µg/mL) or KPE-DMSO control (maximum concentration of 0.05%) in triplicate. Morphological changes post-treatment were observed using a Zeiss Axiovert 200 M inverted microscope (Carl Zeiss, Oberkochen, Germany). Cell viability was assessed using the WST-1 assay. Absorbance was measured at 450 nm with a reference wavelength of 600 nm using a microplate reader, and the data were analyzed with Gen5 software (version 3.12, BioTek, Winooski, VT, USA) [13]. Cell viability was calculated based on absorbance readings at 450 nm, with 600 nm as the reference wavelength;
Cell viability (%) = (A_t_ − A_c_) × 100

A_c_ = absorbance of the control (with DMSO) cells, A_t_ = absorbance of cells treated with different concentrations of KPE [18].

### 2.5. Colony Formation Assay

Melanoma cells were seeded at a density of 3 × 10^5^ cells per well in a 6-well plate and incubated overnight at 37 °C with 5% CO_2_. The following day, the cells were treated with KPE-DCM and KPE-hexane at concentrations of 20 and 50 µg, along with a KPE-DMSO control, for 24 and 48 h. After the treatment period, the cells were harvested using trypsin and reseeded at a density of 4 × 10^2^ cells per well in a 12-well plate for a 10-day incubation. The cells were then washed with PBS, fixed with 100% methanol for 15 min, and stained with 0.5% crystal violet solution for 10 min. Finally, the stained cells were examined under a microscope, and the images were quantified using ImageJ software (Version 1.8.0, National Institute of Health, Bethesda, MD, USA).

### 2.6. TUNEL (TdT-Mediated dUTP Nick-End Labeling) Staining

Following a 48-h treatment of melanoma cells with KPE-DCM and KPE-hexaane, along with a KPE-DMSO control, a TUNEL assay was conducted. This assay was performed according to the manufacturer’s instructions (Promega, Madison, WI, USA). Additionally, the cells were stained with DAPI (4′,6-diamidino-2-phenylindole) using a mounting solution. TUNEL-positive cells and DAPI-stained cells were then examined under ECLIPSE Ts2R inverted microscope (Nikon, Melville, NY, USA), and imaged using the NIS-Elements BR software (Ver 4.00, Nikon) [19].

### 2.7. Wound Healing Assay

Melanoma cells were seeded at a density of 3 × 10^5^ cells per well in 6-well plates and incubated overnight at 37 °C with 5% CO_2_. Once the cells reached 70–80% confluency, a scratch was made in the middle of each well using 200 µL pipette tips. The initial scratch was documented with an Olympus CKX41 microscope (Olympus, Tokyo, Japan). Subsequently, the cells were treated with KPE-DCM and KPE-hexane at concentrations of 20 and 50 µg/mL, along with a KPE-DMSO control, for 24 and 48 h. Images were taken at various time points, and the wound area was analyzed using ImageJ software (Version 1.8.0, National Institute of Health).

### 2.8. Invasion Assay

A transwell invasion assay was performed using a 24-well plate with 8 µm pore membrane inserts (Falcon; Franklin Lakes, NJ, USA). The inserts were coated with 300 µg/mL Matrigel (Corning; Corning, NY, USA) and incubated at 37 °C for 1 h. Melanoma cells were treated with KPE-DCM and KPE-hexane at concentrations of 20 and 50 µg/mL, along with a KPE-DMSO control, for 24 h. Subsequently, 2 × 10^5^ cells in serum-free medium were seeded into the upper chamber of the trans-wells. The lower chamber was filled with 750 µL of culture medium containing 10% FBS and incubated at 37 °C with 5% CO_2_ for 24 h. The following day, the cells were washed, fixed with 4% paraformaldehyde for 5 min, permeabilized with methanol at room temperature (RT) for 20 min, and stained with 0.1% crystal violet solution for 10 min. After removing excess stains and washing with water, the invaded cells were observed under an Olympus CKX41 microscope (Olympus) [20].

### 2.9. Apoptosis Assay by FACs

Melanoma cells (3 × 10^5^ cells/well) were seeded into 6-well plates and allowed to incubate overnight. Subsequently, the cells were pretreated with 20 µM Pan Caspase inhibitor (Z-VED-FMK, R&D Systems, Minneapolis, MN, USA) for 1 h. Following this pretreatment, KPE-DCM and KPE-hexane were added to the respective wells, and the cells were incubated for an additional 48 h. The cells were then washed twice and resuspended in 1 mL of 1 × Annexin V binding buffer. A 100 µL aliquot of this suspension was transferred to a 1.5 mL microtube, to which 5 µL of FITC-Annexin V and 5 µL of PI were added. After a 15-min incubation on ice in the dark, 400 µL of 1 × Annexin V binding buffer was added, and the samples were analyzed by flow cytometry using the NovoCyte 1000 system. Data visualization was performed using NovoExpress software (Version 1.6.0, ACEA Biosciences, San Diego, CA, USA).

### 2.10. Cell Cycle Assay

Melanoma cells (2.5 × 10^5^ cells/well) were plated in six-well plates and incubated overnight. Following this, KPE-DCM and KPE-hexane extracts and KPE-DMSO control were added to the respective wells, and the cells were incubated for 24 and 48 h. The cells were then collected and fixed in 70% ethanol at 4 °C for 1 h. After fixation, the cells were resuspended in DPBS and treated with 100 µg/mL RNase A in PBS for 10 min in the dark. Following this incubation, cells were stained with 50 µg/mL propidium iodide (PI) (BD Life Sciences, San Diego, CA, USA). The samples were kept on ice in the dark until PI fluorescence intensity was measured using flow cytometry on the NovoCyte 1000 system, with data analysis performed using NovoExpress software (ACEA Biosciences).

### 2.11. Mitochondrial Membrane Potential (MMP)

Melanoma cells (2.5 × 10^5^ cells/well) were plated in 6-well plates and incubated overnight. The treatment conditions mirrored those used in the apoptosis assay. Post-treatment, the cells were stained with JC-1 reagent at a final concentration of 2 µM (Enzo Life Sciences, Farmingdale, NY, USA) following standard procedures. After staining, the cells were washed and resuspended in PBS, and their fluorescence was measured using the NovoCyte 1000 flow cytometer. Data visualization was performed using NovoExpress software (ACEA Biosciences).

### 2.12. Reactive Oxygen Species (ROS) Assay by FACs

Melanoma cells (2.5 × 10^5^ cells/well) were seeded into a 6-well plate and incubated overnight. The following day, the cells were pretreated with N-acetyl cysteine (NAC) for 1 h, followed by treatment with KPE-hexane and KPE-DCM for 24 h. After the treatment, the cells were harvested and stained with 2′,7′-Dichlorodihydrofluorescein diacetate (Enzo Life Sciences, Farmingdale, NY, USA). Fluorescence measurements were performed using the NovoCyte 1000 flow cytometer, and the results were visualized using NovoExpress software (ACEA Biosciences).

### 2.13. Western Blot Analysis

Melanoma cells (5 × 10^5^ cells/dish) were plated in 60-mm dishes and incubated overnight. The next day, cells were treated with KPE-DCM at concentrations of 0, 10, and 20 µg/mL, and with KPE-hexane at concentrations of 0, 15, 30, and 60 µg/mL for 24 and 48 h, under conditions of 37 °C and 5% CO_2_. After treatment, the cells were washed once with DPBS, lysed in RIPA buffer (Pierce, Rockford, IL, USA), and stored at −20 °C until further analysis. The total protein concentration was measured using the DC protein assay (Bio-Rad, Hercules, CA, USA). Thirty micrograms of total protein from each sample were subjected to 10–12% SDS-PAGE and transferred onto PVDF membranes. The membranes were blocked with 5% nonfat milk at RT for 1 h and then incubated overnight at 4 °C with primary antibodies (Abs) against beta-actin, caspase-3, caspase-9, Bcl-2, Bax, PARP, and MuD (C22B3). Following primary Ab incubation, the membranes were washed three times with TBS-T and incubated with HRP-conjugated secondary Abs (Goat antimouse and Goat antirabbit) at RT for 2 h. Specific proteins were detected using an enhanced chemiluminescence kit (Bio-Rad, Hercules, CA, USA).

### 2.14. Stable Transfection in A375 and A2058 Cells

A375 and A2058 cells were stably transfected with either the pEGFPC1-MuD plasmid or the control pEGFPC1 empty vector using Lipofectamine 2000 reagent (Invitrogen, Carlsbad, CA). The day after transfection, cells were selected in a primary cell culture medium. After two weeks, individual clones were randomly isolated, and each clone was plated separately. The stable expression of MuD in these cells was confirmed by observing green fluorescence under a fluorescence microscope and through immunoblotting.

### 2.15. Statistical Analysis

Each experiment was performed at least three times independently, and results were presented as mean ± SD. Statistical significance was assessed using the Student’s *t*-test, with *p* < 0.05, *p* < 0.01, or *p* < 0.001 considered significant.

## 3. Results

Figure 2A displays the results of the HPLC characterization of the various KPE extracts. In the case of all KPE extracts the predominant bioactive component that shows up is polymethoxyflavones (PMFs). The extraction of these PMFs depends on the solvent system and the extraction procedure. In their study, Asamenew et al., 2019 [21], reported the successful use of UPLC to identify various KPEs. They reported the presence of various PMFs in KPEs: 6-Hydroxy-7,4′-Dimethoxyflavone,14.23 RT(min); 5,7,3′,4′-Tetramethoxyflavone (tetramethylluteolin), 16.86 RT(min); 3,5,7,3′,4′-Pentamethoxyflavone (pentamethaquercetin), 17.82 RT(min); 5,7-Dimethoxyflavone, 18.32 RT(min); 5,7,4′-Trimethoxyflavone (trimethylapigenin), 18.82 RT(min); 3,5,7-Trimethoxyflavone, 20.12 RT(min); 3,5,7,4′-Tetramethoxyflavone (tetramethylkaempferol) 20.38 RT(min); 5-Hydroxy-7,3′,4′-trimethoxyflavone, 25.11 RT(min); 5-Hydroxy-3,7,3′,4′-tetramethoxyflavone (ayanin), 27.29 RT(min); 5-Hydroxy-7-methoxyfavone (tectochrysin), 27.90 RT(min); 5-Hydroxy-7,4′-dimethoxyfalvone, 28.70 RT(min); 5-Hydroxy-3,7-dimethoxyflavone, 30.99 RT(min) and 5-Hydroxy-3,7,4′-trimethoxyflavone, 31.51 RT(min). In this study, our HPLC analysis verified that several of the previously mentioned PMFs were present in the different KPEs. 3,5,7,3′,4′-Pentamethoxyflavone (pentamethaquercetin), 17.82 RT(min); 5,7-Dimethoxyflavone, 18.32 RT(min); 5,7,4′-Trimethoxyflavone (trimethylapigenin), 18.82 RT(min); 3,5,7-Trimethoxyflavone, 20.12 RT(min); 3,5,7,4′-Tetramethoxyflavone (tetramethylkaempferol) 20.38 RT(min); 5-Hydroxy-7,3′,4′-trimethoxyflavone, 25.11 RT(min); 5-Hydroxy-3,7,3′,4′-tetramethoxyflavone (ayanin), 27.29 RT(min); 5-Hydroxy-7-methoxyfavone (tectochrysin), 27.90 RT(min); 5-Hydroxy-7,4′-dimethoxyfalvone, 28.70 RT(min); 5-Hydroxy-3,7-dimethoxyflavone, 30.99 RT(min) and 5-Hydroxy-3,7,4′-trimethoxyflavone, 31.51 RT(min) PMFs peaks were observed. KPE-EtOAc showed significantly higher extraction of more variants of PMFs (Figure 2B), while KPE-hexane and KPE-DCM possessed high-intensity peaks of 5-Hydroxy-3,7-dimethoxyflavone, 30.99 RT(min) and 5-Hydroxy-3,7,4′-trimethoxyflavone, 31.51 RT(min). This individual spectra are represented in Appendix A.

PMFs recovered from plant materials show a broad spectrum of biological activities [22], including anticarcinogenic properties with low toxicity [23]. The bioactivity of PMFs from citrus fruits is well reported, and Li et al. [24] suggested that the anticancer activity of PMFs was directly proportional to the methoxy groups. According to their report, heptamethoxyflavone exhibited higher anticancer activity than hexamethoxyflavone. KPE is an alternative PMF-rich resource. Nobiletin (5,6,7,8,3′,4′-hexamethoxyflavone) and tangeritin (5,6,7,8,3′-pentamethoxy-flavone) are the most abundant PMFs in Citrus plants, but those in KPEs are structurally different from the ones in citrus plants. The PMFs in KPEs are unique in that only specific positions of flavone structure, such as C-3, C-5, C-7, C-3′ and C-4′, are methoxylated [25].

To investigate the dose- and time-dependent effects of fractionated KPE-MeOH, KPE-EtOAc, KPE-DCM, and KPE-hexane) on A375 and A2058 human melanoma cells, the cells were treated with increasing concentrations of extracts (0–50 µg/mL) for 24 h and 48 h. As shown in Figure 3A,B, and Appendix A, KPE-DCM showed the strongest effect and KPE-hexane showed the lowest effect on both melanoma cell lines among all the fractionated extracts tested. The IC_50_ values of KPE-DCM and KPE-hexane were less than 20 µg/mL and more than 50 µg/mL at 24 h and 48 h, respectively, in both melanoma cell lines (Appendix A). Besides, it was also observed that increasing concentrations of extracts caused abrupt morphological changes (Appendix A). We thus choose two fractions (KPE-DCM; 20 µg/mL and KPE-hexane 50 µg/mL) for all subsequent experiments.

A colony formation assay was performed to examine the ability of a single cell to grow into a colony, and TUNEL staining was conducted to assess the apoptotic and DNA-damaged cells. The results showed that KPE-DCM-treated cells brought about significant inhibition of colony formation compared to the control and KPE-hexane-treated cells (Figure 4A). DNA fragmented cells were also observed following TUNEL staining. As observed from the results, KPE-DCMtreated cells resulted in TUNEL-positive cells (green) in both melanoma cells, but KPE-hexane-treated cells showed a positive effect on A375 cells only and had no impact on control cells. This result suggests that KPE–DCM affects colony formation and DNA-fragmented apoptotic cells, but, in the case of KPE-hexane, it affected DNA-fragmented cells in A375 but had no inhibitory effect on colony formation.

Cell migration and invasion were measured to evaluate whether extracts prevent the metastatic proclivity of melanoma cells. KPE-DCM was found to decrease migration but, KPE-hexane and the DMSO control did not prevent migration in both melanoma cells (Figure 5A,B). In addition, cell invasion significantly decreased after 48 h treatment with KPE-DCM and KPE-hexane with respect to A375 cells. However, this was not the case with A2058 cells (Figure 5C,D). These results suggest that KPE-DCM inhibits melanoma cell migration and invasion.

Cell cycle arrest is one of the most important mechanisms by which anticancer agents exert their inhibitory effects. Therefore, the impact of KPE on cell cycle phase dissemination with respect to melanoma cells via flow cytometry (Figure 6A) was assessed. Our results interestingly confirmed that when the two melanoma cells were treated with KPE-DCM and KPE-hexane for 24 h. As shown in Figure 6B, the cell population of the G_2_ phase was approximately 8.9%, 11.2%, and 28.7%, respectively, after treatment with control, KPE-hexane and KPE-DCM for 24 h in the case of A375 cells. On the other hand, A2058 cells do not influence cell cycle arrest following treatment with KPE-hexane and KPE-DCM. This result suggests that the cell cycle transition from G_0_/G_1_ into the G_2_ phase was promoted by KPE-DCM in A375 cells but had no effect when it came to A2058 cells.

Additionally, we also studied the impact of the extracts on cell cycle marker protein using a western blot assay. As shown in the results, treated cells showed increased p53 and p21 protein levels. According to the results, KPE-DCM impacted cell proliferation by arresting the G2 phase in A375 cells but not in A2058. Moreover, KPE-hexane had no effect on the cell cycle in both melanomas.

To investigate the apoptotic pathway induced by KPE, we performed an Annexin V/PI staining assay in Z-VAD-FMK (pancaspase inhibitor; 20 µM)-pretreated and KPE-DCM and KPE-hexane-treated A375 and A2058 cells. As shown in Figure 7, hexane- and DCM-treated cells increased apoptotic cell population compared to the control without Z-VAD-FMK in both melanoma cells. Conversely, Z-VAD-FMK pretreatment substantially suppressed this effect in both melanoma cells. Additionally, cell viability significantly recovered with Z-VAD-FMK pretreatment. Cell survival recovered following treatment with a pancaspase inhibitor. Based on these results, it is suggested that KPE-hexane and KPE-DCM effectively induced apoptosis pathways in both melanoma cells.

To investigate the regulation of MMP (Δ*Ψ*m) and the evolution of cellular ROS following interaction with KPE-hexane and KPE-DCM, both melanoma cells were stained with JC-1 (MMP) and DCFH-DA (ROS). As shown in Figure 8A. KPE-hexane and KPE-DCM treated cells showed loss of mitochondrial membrane potential compared to control in both melanoma cells (FITC-positive population at extracts treated cells >50%, control approximately 10%). Moreover, KPE-DCM induced ROS production, NAC (0.5 mM) pretreatment diminished ROS production and KPE-hexane had no effect on A375 cells. On the other hand, in KPE-hexane and KPE-DCM-treated A2058 cells, no significant increase in ROS production was observed. In order to confirm that the KPE-hexane and KPE-DCM cytotoxic effects are due to ROS accumulation, NAC pretreating cells were used; it was observed that suppressed KPE-DCM-induced cell death in both melanoma cells. These investigations confirm that KPE-DCM treatment and ROS increase correlation with mitochondrial depolarization.

We performed western blotting to examine the effect of KPE-hexane and KPE-DCM on the activation of pro-apoptotic protein (cleavage-PARP, cleavage-caspase-3 and Bax) and anti-apoptotic (MuD and Bcl-2) protein inactivation. As shown in Figure 9, KPE-hexane and KPE-DCM treatment upregulated cleavage-PARP, cleavage-caspase-3, and Bax. As expected, they downregulated the expression levels of anti-apoptotic proteins, which are MuD and Bcl-2 in A375 and A2058. These results are suggestive of the effect of KPE-hexane and KPE-DCM on the upregulation of the apoptotic cell death signaling pathway.

## 4. Discussion

Surgery, radiotherapy, chemotherapy, immunotherapy, and various biological agents are currently in use to treat malignant melanoma [26]. Exposure to UV radiation is one of the most important risk factors for “malignant melanoma” or “skin cancer” leading to 80% death. Despite this arsenal of treatments, the prognosis is poor. In the present study, we have demonstrated that KPEs induce apoptotic cell death in melanoma cells. It has potent cytotoxic effects against various cancer cell lines [14,27].

KPE exhibits no effects on NIH-3T3 preadipocytes even at the highest concentration of 250 µg/mL [28]. However, concentrations exceeding 62.5 µg/mL induce cytotoxicity in the mouse macrophage-like cell line RAW264.7 and the human keratinocyte cell line HaCaT [29]. Furthermore, KPE-derived extracts demonstrate varying effects on mouse microglial BV-2 cells: KP1 (ethanolic) shows effects at 40 µg/mL, KP2-3 (hexane, chloroform) at 200 µg/mL, and KP4-5 (ethyl acetate, residue) at 1000 µg/mL [30]. In this present study, the tested effects of KPE focused on the decline of the survival of human melanoma cells as observed based on the results obtained from testing the two melanoma (A375 and A2058) cells. The results confirmed that both the melanoma cells tested were inhibited by KPE.

Earlier researchers have confirmed the inhibitory effect of KP on migration in various cells [1,12,31]. Our results have shown similarity with previous reports, where KPE-DCM and KPE-hexane were found to inhibit cell migration in a time-dependent manner as defined through wound healing assays in two melanoma cell lines (Figure 5A,B). Furthermore, we used a transwell assay to confirm the migratory function. KPE-DCM significantly suppressed migration, whereas KPE-hexane did not affect migration. Additionally, KPE-DCM similarly inhibited invasion, but KPE-hexane had no effect on the two melanoma cell lines experimented on (Figure 5C,D).

Moreover, caspase-dependent apoptosis is a known type of programmed cell death that has been frequently studied [32]. We observed that KPE enhanced apoptotic cells in the absence and presence of pancaspase inhibitor Z-VAD-FMK. Interestingly, these apoptotic cells were significantly suppressed by the caspase inhibitor (Figure 7). Moreover, we observed MMP and intracellular ROS levels that were suggestive that KPE brought about mitochondrial-mediated apoptosis in melanoma cells, as shown in Figure 8. KPE treatment increased MMP levels and upregulated the generation of ROS in A375 cells but not in A2058 cells. However, pretreatment with a ROS scavenger (NAC) suppressed KPE-induced cell death and ROS generation. These results clearly pointed out the KPE-induced mitochondrial-mediated apoptotic cell death as the mechanism behind the inhibition of the melanoma cells [33,34].

The mitochondria-mediated apoptosis is regulated by a delicate balance between the opposing actions of pro-apoptotic and anti-apoptotic Bcl-2 family members. The mitochondrial/caspase-mediated signaling cascade is a major apoptotic pathway that is characterized by MMP, whose disruption leads to the release of cytochrome c into cytoplasm, triggering the caspase cascade and PARP cleavage [14,35]. In this study, KPE treatment resulted in the upregulation of cleavage PARP and caspase-3 and downregulation of Bcl-2 and anti-apoptotic protein MuD expression in both cells (Figure 10). It was previously reported that TMF effectively inhibited the proliferation of SNU-16 human gastric cancer cells in a concentration-dependent manner by inducing apoptosis through various pathways, including ER stress, as evidenced by multiple cellular markers among the DMF, TMF, and PMF isolated from brewed tea samples [12].

The black ginger rhizome contains several PMFs [36,37,38,39] with high anti-inflammatory activity [3,39,40]. A previous study highlighted three flavonoids—5,7-dimethoxyflavone (DMF), 5,7,4′-trimethoxyflavone (TMF), and 3,5,7,3′,4′-pentamethoxyflavone (PMF)—that are major compounds in black ginger [38]. DMF, one of these major flavonoids, has been mentioned in several previous studies. As a potent anti-inflammatory agent, it can inhibit degranulation. In addition, studies have indicated its ability to inhibit tumor necrosis factor-α, interleukin-4, and monocyte chemoattractant protein-1 production [3,39] as well as nitric oxide release [40].

## 5. Conclusions

For the first time, we have confirmed that Laos KPE exhibits anticancer effects on human melanoma A375 and A2058 cell lines. We confirmed that KPE-DCM is more effective than KPE-hexane in anti-proliferation, inhibition of migration, and invasion assays. Our hypothesis is that KPE-hexane did not lead to positive extraction of the principal anticancer compound 3,5,7,3′,4′-pentamethoxyflavone. Notably, the study found that KPE-DCM and KPE-hexane inhibited melanoma through the ROS-mediated apoptotic cell death mechanism. KPE affected the upregulation of cleavage PARP and caspase-3 and the downregulation of Bcl-2 and anti-apoptotic protein MuD expression in both cells. More investigations on other cell death pathways and key regulatory proteins involved in KPE-induced cell death will be needed to arrive at conclusions. Moreover, exploring the role of specific ROS sources and their interaction with mitochondrial dynamics could provide a deeper understanding of how KPE induces apoptosis.

## Figures and Tables

**Figure 1 medicina-60-01371-f001:**
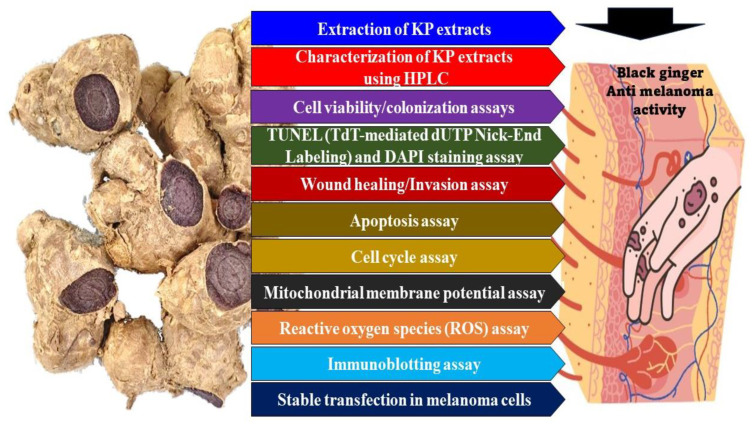
Schematic of the workflow followed in this study.

**Figure 2 medicina-60-01371-f002:**
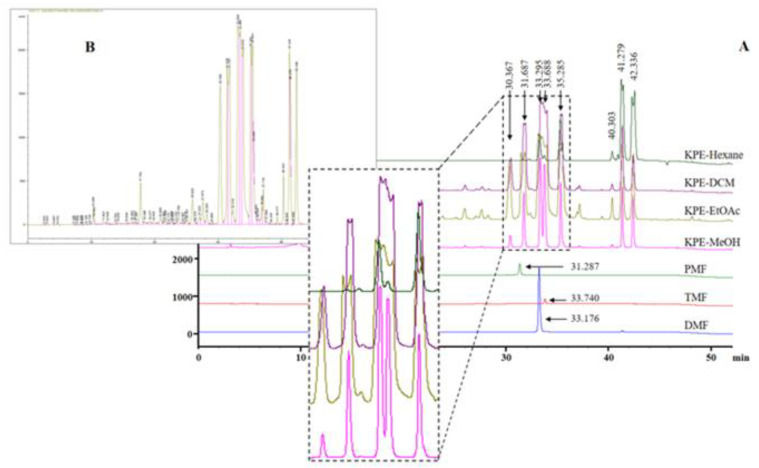
(**A**) HPLC results of KPE bioactive compounds; Inset (**B**) showing PMF peaks in KPE-EtOAc.

**Figure 3 medicina-60-01371-f003:**
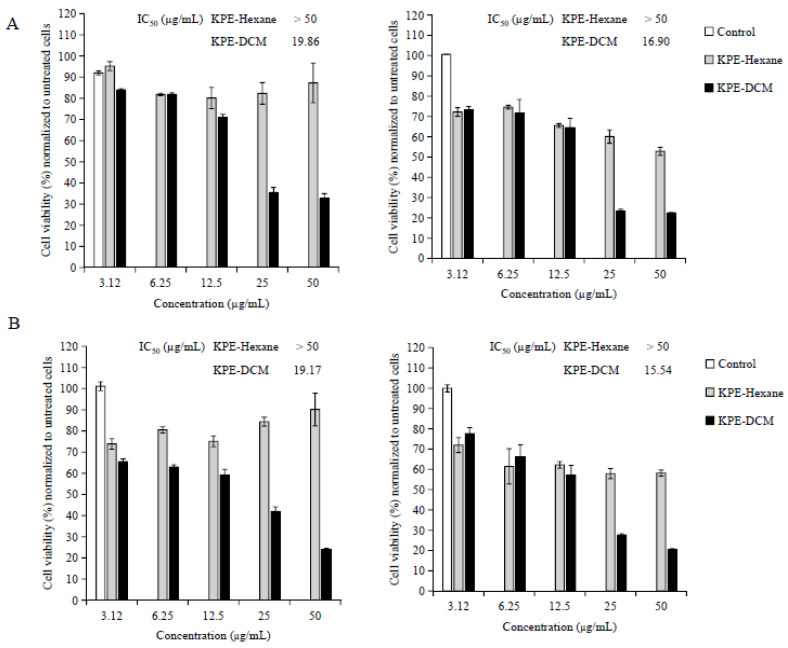
The cytotoxic effects of black ginger extracts against A375 and A2058 melanoma cells. Cell viability was measured using a WST-1 assay. (**A**) A375 cells (**B**) A2058 cells; 24 h (left) and 48 h (right). The data were presented as mean ± SD of two independent experiments.

**Figure 4 medicina-60-01371-f004:**
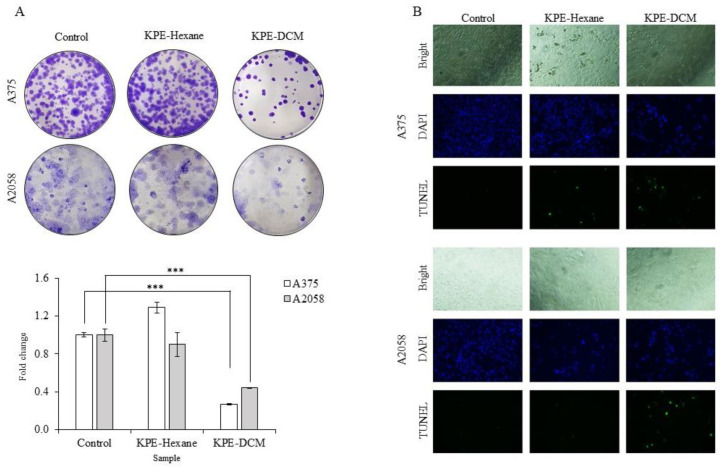
The effect of black ginger extracts on colony formation and colorimetric assay on A375 and A2058 human melanoma cells. (**A**) Cells were treated with KPE-DCM; 20 µg/mL, KPE-hexane; 50 µg/mL for 24 h and 48 h and reseeded into 12-well plates. After a 10-day incubation period, colonies were stained with 0.5% crystal violet. Colony formation assay showing colony numbers per well along with graphical representation. (**B**) After treatment for 48 h, cells were stained with TUNEL. Cells were observed under the fluorescence microscope (×200). Experiments were performed in three independent biological replicas and showed representative image data (*** *p* < 0.001).

**Figure 5 medicina-60-01371-f005:**
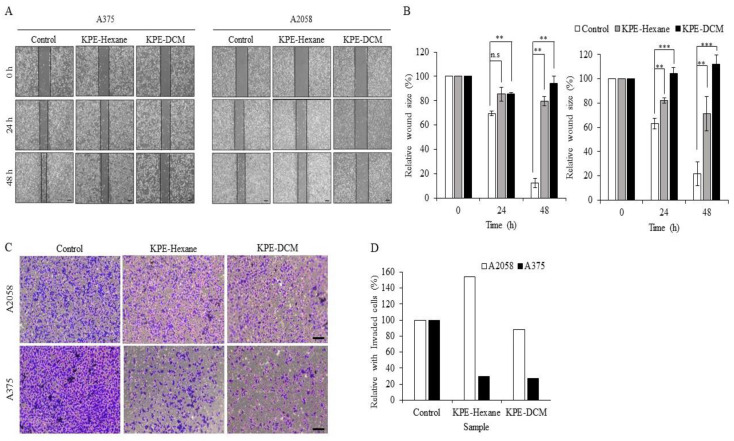
Effects of black ginger extracts of metastatic on A375 and A2058 cells. (**A**) Migration representative photographic result after treatment with KPE-DCM; 20 µg/mL, KPE-hexane; 50 µg/mL for 24 h and 48 h. (**B**) Migrated quantitative graph. (**C**) Invaded representative photographic result after being treated with KPE-DCM; 20 µg/mL, KPE-hexane; 50 µg/mL for 48 h. (**D**) invaded quantitative graph. Quantitative data analysis using ImageJ software 1.54. The data were presented as mean ± SD of three independent experiments ** *p* < 0.01, *** *p* < 0.001).

**Figure 6 medicina-60-01371-f006:**
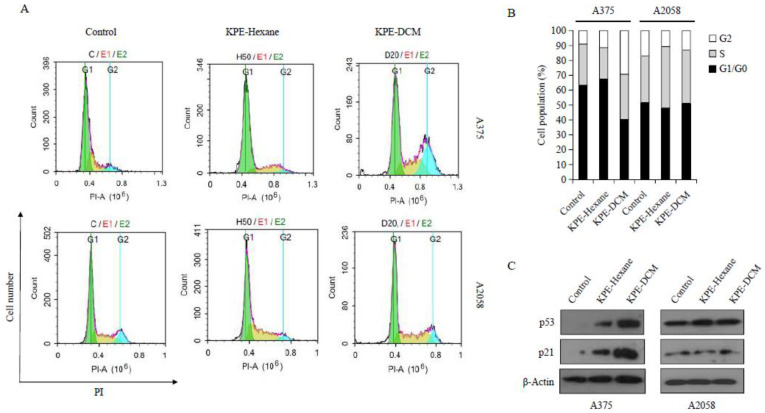
Black ginger extracts were induced (which phase) on A375 and A2058 cells. (**A**) The representative plot shows cells in the G_0_/G_1_ phase, S phase, and G_2_ phase. (**B**) The graph shows the percentage of the cell population. (**C**) The effect of black ginger extracts on the expression level of cell cycle regulatory proteins was examined by a western blot assay. The data were presented as mean ± SD of three independent experiments.

**Figure 7 medicina-60-01371-f007:**
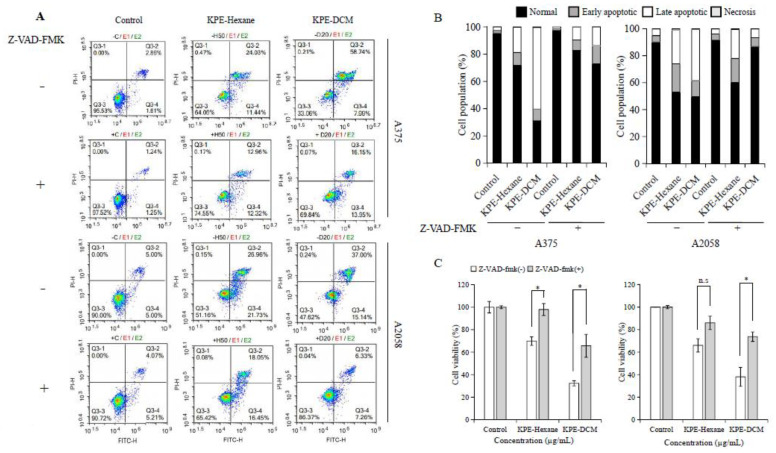
Black ginger induces apoptosis cell population in A375 and A2058 cells. The cells were pretreated with and without Z-VAD-FMK (pancaspase inhibitors) for 1 h. Then, treatment DCM; 20 µg/mL, Hexane; 50 µg/mL for 48 h. (**A**) cells were determined by flow cytometry after staining with Annexin V/Propidium iodide (PI). (**B**) percentage of the population is shown as a graph. (**C**) Cell viability assay. The data were presented as mean ± SD of three independent experiments (* *p* < 0.05).

**Figure 8 medicina-60-01371-f008:**
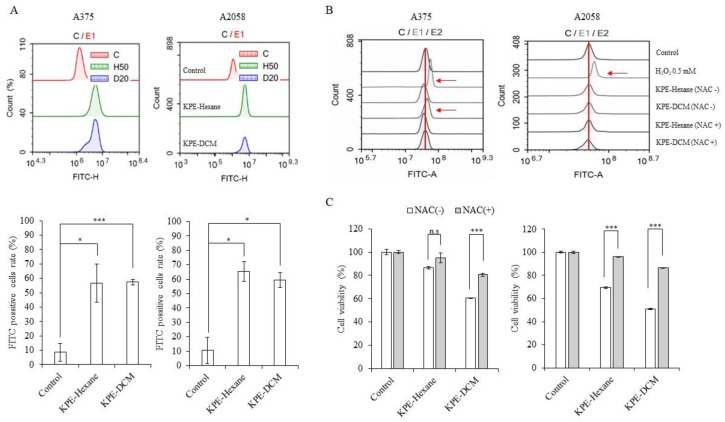
The decreased mitochondrial membrane potential in melanoma cells. The cells were treated with KPE-hexane and KPE-DCM for 24 h. (**A**) Mitochondrial membrane potential assay measured under flow cytometry and representative bar graph. (**B**) ROS assay after KPE-hexane and KPE-DCM treatment absence and presence of N-acetyl-L-cysteine; 1 mM for 24 h. and (**C**) Cell viability assay. The data were presented as mean ± SD of three independent experiments (* *p* < 0.05, *** *p* < 0.001).

**Figure 9 medicina-60-01371-f009:**
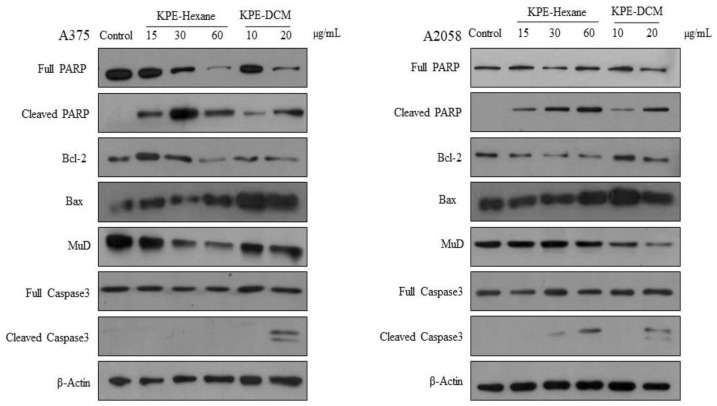
Effect of black ginger extracts on apoptosis-related proteins in both melanoma cells. The cells were treated with KPE-hexane and KPE-DCM for 48 h. Representative western blots of PARP, Bcl-2, Bax, MuD, Caspase-3 and β-actin. Experiments were performed in three independent biological replicas and showed representative image data.

**Figure 10 medicina-60-01371-f010:**
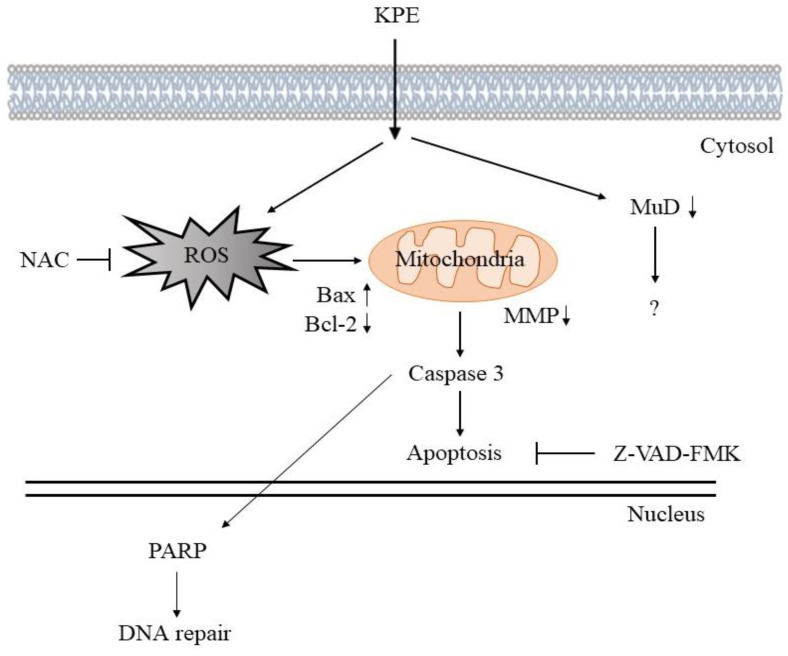
The speculated mechanism behind the anticancer effect of the KPEs.

## Data Availability

Data will be made available on request.

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
