# Peer review of "Evaluating the Diverse Anticancer Effects of Laos Kaempferia parviflora (Black Ginger) on Human Melanoma Cell Lines"

_medicina, 2024, doi:10.3390/medicina60081371_

Round 1
Reviewer 1 Report
Comments and Suggestions for Authors
Please see attach file.

Author Response
Reviewer 1: Please see attach file.
Answer: We would like to thank the reviewer for his time and efforts to improve our manuscript. We have received your comments and have revised the manuscript based on your comments. Format checked and reference (in discussion) added as per your suggestion We have highlighted the changes made to the manuscript using track changes. We have expanded on the conclusion. We greatly appreciate the opportunity we have been given to revise the manuscript. Thank you again.
Reviewer 2 Report
Comments and Suggestions for Authors
The manuscript by Jae-Wook Oh and co-workers, reports the effects of Kaempferia parviflora (KP) (black ginger) extracts on human melanoma cell lines, A375 and A2058.
The manuscript shows the performance of very interesting experiments and the writing is mostly structured. I would only recommend the following to the authors to improve their research:
1.- It is advisable to include the most relevant data in the abstract.
2.- In the methodology it is advisable to put the references on which they based their experiments.
3.-It is advisable to report how the extracts was prepared and manipulated for their administration in the biological trials. What was the vehicle used to administer the extracts? If an organic solvent was used, it is recommended to report the maximum concentration that was used and the effect of the vehicle in assays. How did they take care of the sterility of the extracts studied?
4.-It is advisable to include the vehicle control in the assays.
5.-It is advisable to use a positive control (ej. drug) in the assays.
6.-In the chromatogram (Figure 2A) it is advisable to include the retention times of the standard compounds (bioactive compounds). It is also advisable to make peak expansions in the chromatograms of the extracts that contain the bioactive compounds.
7.- It is also highly recommended to report which specific standard compounds were used in the HPLC analyses. Also, include full chromatograms of the extracts and standard compounds in the supplementary material so that readers can clearly compare.
8.-Line 225: ….In this study, our HPLC results confirmed the presence of few of the above PMFs in the various KPEs…. Did the authors use the same conditions as in the cited reference? For example, mobile phase, chromatography column, same chromatographic technique, temperature, flow, pressure, etc.
9.- IC50, please put the number 50 as a subscript in figures, tables and paragraphs.
10.- Line 263: ….We thus choose two fractions (KPE-DCM; 20 μg/mL and KPE-hexane 50 μg/mL) for all subsequent experiments.…. What was the selection criterion? Why was it decided not to perform subsequent bioassays with the ethyl acetate extract? According to the supplementary material KPE-DCM and KPE- Eto-Aco were the most active in the cell viabillity assay.
11.- It is recommended to improve the discussion and conclusions. For example, in the discussion you can further contrast your data with information from plant extracts that contain similar secondary metabolites and that have been tested for similar bioactivity, trying to highlight the data found in your experiments.
Author Response
Reviewer 2:
The manuscript by Jae-Wook Oh and co-workers, reports the effects of Kaempferia parviflora (KP) (black ginger) extracts on human melanoma cell lines, A375 and A2058.
The manuscript shows the performance of very interesting experiments and the writing is mostly structured. I would only recommend the following to the authors to improve their research:
1.- It is advisable to include the most relevant data in the abstract.
Answer: We thank you for you kind and encouraging words. We have revised the abstract according to your comments. Thank you.
2.- In the methodology it is advisable to put the references on which they based their experiments.
Answer: We have added references as per your suggestion and we have also modified the references correcting about 15 incorrect citations, we have proof read the entire reference part once again. Thank you.
3.-It is advisable to report how the extracts was prepared and manipulated for their administration in the biological trials. What was the vehicle used to administer the extracts? If an organic solvent was used, it is recommended to report the maximum concentration that was used and the effect of the vehicle in assays. How did they take care of the sterility of the extracts studied?
Answer: as follows your comments, we added the preparation of samples for biological experiment in section 2.3 cell culture and sample preparation:“KPE-extracts (methanol, hexane, dichloromethane, EtoAc) were stored in a deep freezer until use. KPE’s dissolved in dimethyl sulfoxide (DMSO) at a concentration of 100 mg/mL was used for in vitro biological tests .”
Thank you
4.-It is advisable to include the vehicle control in the assays.
Answer: Included and mentioned in the manuscript. Thank you.
5.-It is advisable to use a positive control (ej. drug) in the assays.
Answer: We will consider this in our future experiments, as of now we go with our conducted experimental set up, thank you for your kind understanding.
6.-In the chromatogram (Figure 2A) it is advisable to include the retention times of the standard compounds (bioactive compounds). It is also advisable to make peak expansions in the chromatograms of the extracts that contain the bioactive compounds.
Answer: Please check Figure 2A. we have included the retention times of the standard compounds and peak expansions, thank you.
7.- It is also highly recommended to report which specific standard compounds were used in the HPLC analyses. Also, include full chromatograms of the extracts and standard compounds in the supplementary material so that readers can clearly compare.
Answer: Please check the Supplementary figure. we have now provided the full chromatogram of the extracts and standard compounds as Supporting Figure. .
8.-Line 225: ….In this study, our HPLC results confirmed the presence of few of the above PMFs in the various KPEs…. Did the authors use the same conditions as in the cited reference? For example, mobile phase, chromatography column, same chromatographic technique, temperature, flow, pressure, etc.
Answer: The HPLC conditions were nearly the same as in the reference, with minor modifications based on our analytical expertise. Thank you.
9.- IC50, please put the number 50 as a subscript in figures, tables and paragraphs.
Answer: Modified. Thank you. Sorry about that.
10.- Line 263: ….We thus choose two fractions (KPE-DCM; 20 μg/mL and KPE-hexane 50 μg/mL) for all subsequent experiments.…. What was the selection criterion? Why was it decided not to perform subsequent bioassays with the ethyl acetate extract? According to the supplementary material KPE-DCM and KPE- Eto-Aco were the most active in the cell viabillity assay.
Answer: In oder to stream line the process and cut down on wastages of resources for running multiple variables, we restricted to selecting KPE-DCM and KPE-hexane based on the most effective substance and the least effective substance on melanoma. Thank you.
11.- It is recommended to improve the discussion and conclusions. For example, in the discussion you can further contrast your data with information from plant extracts that contain similar secondary metabolites and that have been tested for similar bioactivity, trying to highlight the data found in your experiments.
Answer: Revised as per your suggestion. Thank you.
Reviewer 3 Report
Comments and Suggestions for Authors
The paper “Evaluating the diverse anti-cancer effects of Laos Kaempferia parviflora (black ginger) on human melanoma cell lines” describes the anti-melanoma potential of Laos Kaempferia parviflora extracts in human cell lines. The manuscript seems to be interesting and actual, but some points should be clarified. It seems that this manuscript is more applicable for the Plants journal. In Abstract: the authors mentioned that Kaempferia parviflora (KP) extracts in methanol, DCM, ethyl acetate and hexane were tested against two melanoma cell lines. But only KPE-Hexane and KPE-DCM were studied. It should be performed in the Abstract and Conclusions parts. In Introduction: the authors should add information on the main components’ of KP bioactivity, including methoxyflavones. IC50 should be performed as µM. For better understanding, the authors should perform the composition of the extract in the form of a Table: the name of the compound and its structure, extract/compound/time RT/reference, if there is a coincidence with a literature data. Has the quantitative analysis of the extracted compounds been evaluated? Have the authors investigated the toxicity of extracts against normal cells (toxicity)? Fig. 6,7: the quality of this figures should be improved, it’s unreadable. Conclusions should be written in more detail, updating information about KPE-Hexane и KPE-DCM.
Author Response
Reviewer 3:
The paper “Evaluating the diverse anti-cancer effects of Laos Kaempferia parviflora (black ginger) on human melanoma cell lines” describes the anti-melanoma potential of Laos Kaempferia parviflora extracts in human cell lines. The manuscript seems to be interesting and actual, but some points should be clarified.
- 1.It seems that this manuscript is more applicable for the Plants journal.
Answer: Although, the plant derived extract is from a plant, the medicinal aspect of the plant has been discussed and hence Medicina will be a appropriate home for our manuscript. We thank you for your consideration.
- 2. In Abstract: the authors mentioned that Kaempferia parviflora (KP) extracts in methanol, DCM, ethyl acetate and hexane were tested against two melanoma cell lines. But only KPE-Hexane and KPE-DCM were studied. It should be performed in the Abstract and Conclusions parts.
Answer: We modified line 18, thank you for guiding us.
- In Introduction: the authors should add information on the main components’ of KP bioactivity, including methoxyflavones.
Answer: Added. Thank you.
- 4.IC50 should be performed as µM.
Answer: It is not possible to calculate the molar concentration of the exact concentration because it is a crude extract, not pure compound, thank you for your understanding.
- For better understanding, the authors should perform the composition of the extract in the form of a Table: the name of the compound and its structure, extract/compound/time RT/reference, if there is a coincidence with a literature data.
Answer: The highlight of this paper is to focus on the anticancer aspect of the extract, numerous studies have already highlighted and presented such tables on the analytical identification and characterization of KPE, this is why we have not elaborated as a table but have mentioned its significance in our text. Thank you.
- Has the quantitative analysis of the extracted compounds been evaluated?
Answer: No quantitative analysis was executed. Thank you.
- Have the authors investigated the toxicity of extracts against normal cells (toxicity)?
Answer: Since we were experimenting with a, edible natural plant based compound that is already used for human consumption, we did not check it out against normal cells. Thank you.
- Fig. 6,7: the quality of this figures should be improved, it’s unreadable.
Answer: Figure 6 and 7 have been replaced with clear versions.
- Conclusions should be written in more detail, updating information about KPE-Hexane и KPE-DCM.
Answer: Rewritten. Thank you very much.
Round 2
Reviewer 2 Report
Comments and Suggestions for Authors
The manuscript by Jae-Wook Oh and co-workers, reports the effects of Kaempferia parviflora (KP) (black ginger) extracts on human melanoma 55 cell lines, A375 and A2058.
The authors carefully included changes in the improved version of the manuscript, which satisfy the suggestions made. It only remains to consider the following:
1.- It is recommended to report the DMSO´s maximum concentration and it´s effect in the assays.
2.- It is advisable to include a positive control in the biological assays (drug or active compound previously reported in this kind of experiments). The recommendation is due to the fact that both the 100% viability control and the vehicle control are as important as the positive control. The latter because confirm the correct performance of the experiments and supports the results found in new studies.
3.- It is also highly recommended to report which specific standard compounds were used in the HPLC analyses. Please, clarify what chemical compounds are DMF, TMF and PMF.
4.-Line 261, It is advisable to rewrite the following: ….In this study, our HPLC results confirmed the presence of few of the above PMFs in the various KPEs…., because the authors did not use identical conditions as in the reference.
Good luck
Author Response
Evaluating the diverse anti-cancer effects of Laos Kaempferia parviflora (black ginger) on human melanoma cell lines
We would like to thank the reviewer 2 for his time and efforts to improve our manuscript. We have received your comments and have revised the manuscript based on your comments. We have highlighted the changes made to the manuscript using red color. We greatly appreciate the opportunity we have been given to revise the manuscript.
Thank you again.
==============================
Reviewer 2: Review Report (Round 2)
The manuscript by Jae-Wook Oh and co-workers, reports the effects of Kaempferia parviflora (KP) (black ginger) extracts on human melanoma cell lines, A375 and A2058.
The authors carefully included changes in the improved version of the manuscript, which satisfy the suggestions made.
Answer: Thank you for giving us a positive evaluation.
It only remains to consider the following:
1.- It is recommended to report the DMSO´s maximum concentration and it´s effect in the assays.
Answer: Yes. We mentioned in Method 2.4 (lines 121-122) that the highest concentration of DMSO used was 0.05%. This is a suitable choice, as 0.05% DMSO is known to have no effects on the cells.
2.- It is advisable to include a positive control in the biological assays (drug or active compound previously reported in this kind of experiments). The recommendation is due to the fact that both the 100% viability control and the vehicle control are as important as the positive control. The latter because confirm the correct performance of the experiments and supports the results found in new studies.
Answer: We have referred to a previous study that confirmed KPE's non-toxicity in non-cancerous cells (Line 482-487; as discussed in the Discussion section), also since the source is a natural extract, we were sort of light on it. We thank you for your understanding. Thank you again.
3.- It is also highly recommended to report which specific standard compounds were used in the HPLC analyses. Please, clarify what chemical compounds are DMF, TMF and PMF.
Answer: Please find the updated methods section (2-2, Line105-109), which includes the clarification of the chemical compounds information as per your recommendation.
4.-Line 261, It is advisable to rewrite the following: ….In this study, our HPLC results confirmed the presence of few of the above PMFs in the various KPEs…., because the authors did not use identical conditions as in the reference.
Answer: We have made some revisions as shown below.
: “In this study, our HPLC results confirmed the presence of few of the above PMFs in the various KPEs.” ----> Line 266-267: “In this study, our HPLC analysis verified that several of the previously mentioned PMFs were present in the different KPEs.”
We once again thank you for your time an efforts your valuable comments he no doubt enhanced the quality of our manuscript. Thank you.